# Functioning and Cognition of Portuguese Older Adults Attending in Residential Homes and Day Centers: A Comparative Study

**DOI:** 10.3390/ijerph18137030

**Published:** 2021-06-30

**Authors:** Manuel José Lopes, Lara Guedes de Pinho, César Fonseca, Margarida Goes, Henrique Oliveira, José Garcia-Alonso, Anabela Afonso

**Affiliations:** 1Escola Superior de Enfermagem de São João de Deus, Universidade de Évora, 7000-801 Évora, Portugal; cfonseca@uevora.pt; 2Comprehensive Health Research Centre (CHRC), Universidade de Évora, 7000-801 Évora, Portugal; margarida.goes@ipbeja.pt (M.G.); hjmo@lx.it.pt (H.O.); 3Escola Superior de Saúde, Instituto Politécnico de Beja, 7800-111 Beja, Portugal; 4Instituto de Telecomunicações, 1049-001 Lisboa, Portugal; 5Department of Computer Systems and Telematics Engineering, University of Extremadura, 10003 Cáceres, Spain; jgaralo@unex.es; 6Departamento de Matemática, Escola de Ciências e Tecnologia, Universidade de Évora, 7000-671 Évora, Portugal; aafonso@uevora.pt; 7Centro de Investigação em Matemática e Aplicações, Universidade de Évora, 7000-671 Évora, Portugal

**Keywords:** older adults, functioning, cognition, disability, long-term care, nursing homes, Day Center

## Abstract

The functioning and cognition of older adults can be influenced by different care contexts. We aimed to compare the functioning profiles and cognition of institutionalized and noninstitutionalized older adults and to evaluate the effect of sociodemographic factors on the functioning and cognition. This is a cross-sectional study that included 593 older adults. The data were collected using the Elderly Nursing Core Set and Mini Mental State Examination. Women, older adults who did not attend school and those live in Residential Homes are more likely to have a higher degree of cognitive impairment than men, those who attended school and those frequent Day Centre. The chances of an older adult with moderate or severe cognitive impairment increases with age. Older women, older adults who did not attend school, and older adults who live in Residential Homes had a higher degree of functional problem than men, those who attended school and those who frequent a Day Centre, independently to age. It is necessary to promote the health literacy of older adults throughout life. The implementation of social and health responses should allow older adults to remain in their homes, given the influence of functioning and cognition on self-care and quality of life.

## 1. Introduction

Various international organizations (the World Health Organization (WHO), United Nations (UN), World Bank (WB), Organization for Economic Cooperation and Development (OECD) and the European Commission (EC)) have demonstrated the magnitude of population aging and noted that it is particularly notable in more developed countries. The National Statistical Institute of Portugal (INE), in turn, has examined the issue in Portugal and predicts that the phenomenon of population aging, which is already very significant, will continue to worsen over the next 30 to 40 years [1] and will be characterized mainly by (i) the decrease in mortality in the young population; (ii) increased average life expectancy; and (iii) the replacement of communicable diseases with chronic diseases. Portugal is already the European Union country with the third highest rate of population aging among residents [2].

Although the aging process is inevitable, epidemiological studies clearly show that diseases and functional disabilities, which are often associated with aging, are not inevitable [3]. However, it has been found that the increase in average life expectancy does not necessarily equate to an increase in years lived in good health. From the perspective of life cycle epidemiology [4,5,6], aging is largely determined by the lifestyles one adopts throughout life. In addition to individual factors, indirect factors—such as organizational, community and political factors that support individual behavior—also contribute to these lifestyles [7], and the social environment has a predominant role [3]. Thus, lifestyle is associated with the physiological changes that are characteristic of aging and often create conditions conducive to the development of multimorbidity. Multimorbidity, in turn, is often associated with the development of functional limitations, which may result in dependence for self-care and the need for some degree of supplementation or substitution through health and social care services. Dependence can result from the interaction between a person health condition and their context, i.e., multimorbidity itself does not necessarily mean that a person will become dependent for his/her self-care, and it depends on the context in which the individual exists in terms of both environmental and social resources, as well as his/her relational level.

According to the UN, approximately 46% of the world population aged 60 years or older has disabilities, and 250 million people in that age group experience moderate to severe functional problems [8]. In turn, the empirical evidence [9,10,11] seems to demonstrate that the years lived with dependence have increased. Regarding Portugal, a study developed in the context of primary health care [12] found that 72.7% of the Portuguese adult population has multimorbidity (two or more chronic diseases), and that percentage increases as age advances. In another study, [13] based on the National Health Survey with Physical Examination (INSEF, for its acronym in Portuguese), the prevalence of multimorbidity in the Portuguese population was 38.3%. A study realized with older adults attending institutions for the elderly in Portugal, showed that 68.2% presented multimorbidity [14]. In turn, regarding dependence for self-care, a set of studies developed in several municipalities of the country suggests that there are approximately 110,000 people who are dependent for self-care, approximately 48,000 of whom are highly dependent [15].

Multimorbidity and dependence are thus a complex combination of genetic, physiological, environmental, and behavioral factors [16] that have important implications for the provision of care [17]. In turn, health care aimed at people with multimorbidity produces better results when it is structured according to a previous evaluation of functioning [17,18].

Functional capacity is considered an important indicator of the health of the older adults, since its decline brings a risk of loss of autonomy and independence, especially at the level of self-care, which directly affects the quality of life of affected individuals [19]. However, dependence for self-care arises when an individual’s functional capacity decreases to the point that they need support to perform activities of daily living and/or instrumental activities of daily living due to either physical or psychosocial dependence. People who are dependent on self-care should remain in their homes [20], which gives meaning to aging in the home and community context [21].

Thus, the phenomenon of demographic aging requires the adoption of a strategy with two complementary vectors: One focused on promoting active and healthy aging, and the other focused on increasing long-term care [22].

The WHO developed several tools in its creation of an international standardized system of health data. Of particular note is the International Classification of Functioning, Disability and Health—ICF [23], which is used to evaluate the results of health interventions. In addition, as an internationally applicable instrument, it allows national and international comparisons [24]. The ICF includes biological, psychological, social, and environmental factors because it classifies (i) the functioning, disability and health of the person and interrelates those factors with the individual’s health status; (ii) body functions and structures; (iii) activity, which includes the execution of individual tasks or actions; (iv) participation, i.e., involvement in real-life situations; and (v) context, i.e., environmental and personal factors that can act as barriers or facilitators [25,26,27]. In addition to the ICF assessment, it is also important an international comorbidity education aims at better management of this increasing problem [28].

The aim of this study was to compare the functioning profiles and cognition of institutionalized (residential homes) and noninstitutionalized (day centers) older adults and to evaluate the effect of sociodemographic factors on the functioning and cognition.

## 2. Materials and Methods

This is a cross-sectional study and involved a sample of 586 older adults divided into two groups: the Day Centers group (DC) was composed of non-institutionalized older adults living in their homes and attending day centers (N = 102); and the Residential Homes group (RH) was composed of institutionalized older adults living in residential homes (N = 484). The inclusion criteria were as follows: 65 years of age or older; able to sign the informed consent form or have a legal representative available to do so; and freely agreeing to participation in the study.

Data were collected between July 2019 and March 2020 at 18 social support institutions for older adults, which comprised residential homes and day care centers, from the north to the south of mainland Portugal. The institutions were randomly selected, each assigned a number, and an online lottery was conducted. All older adults from the selected institutions who met the criteria were included.

The data were collected using the Elderly Nursing Core Set (ENCS) and Mini Mental State Examination. All researchers/collaborators involved in the study received prior training on how to conduct the collect data in addition to providing all necessary clarifications regarding the contents of the instrument used. Informed consent was obtained from all participants or his/her legal represent. During this period, the document was read in its entirety by the respondent himself/herself or by the health professional if needed to avoid embarrassment to the respondent (for example, if the respondent did not know how to read). Information on the study objectives was provided in full to the respondents and/or their families, and they were informed of the confidentiality of the data. The data were collected using the MIAPE platform [29].

### 2.1. Instruments

Elderly Nursing Core Set (ENCS): The instrument used to assess functioning, the ENCS, was developed by Lopes and Fonseca [30] and it have good psychometrics characteristics, with a Cronbach’s alpha of 0.96 [31]. The ENCS consists of a first section that collects sociodemographic data (age, sex, marital status, education and medical diagnosis), followed by 25 questions based on the ICF and categorized on a Likert scale from 1 to 5 points. The ENCS features four domains, these being: self-care, learning and mental functions, communication and social relationships [31]. The lower the total score the better the functioning [31].

Mini Mental State Examination (MMSE): MMSE was developed by Folstein and collaborators in 1975 and was used to evaluated cognition. It consists of six groups of questions that assess temporal and spatial orientation, memory, attention and calculation, recall, language and constructive skills [32]. The lower the value of the scale, the further the cognitive degradation.

### 2.2. Ethical Considerations

The study was approved by the Ethics Committee for Scientific Research in the Areas of Human Health and Welfare of the University of Évora (reference number 19013). All the recommendations of the Declaration of Helsinki of 1964 and its subsequent amendments was followed [33].

### 2.3. Data Analysis

A summary statistical analysis was performed to describe the sociodemographic variables and scores for the overall functional profile of the sample elements using absolute and relative frequencies. The Mann–Whitney U test was used to compare MMSE scores and dependency levels by groups. A chi-square test of independence was performed to examine the relation between group and the severity level of dependence. The logistic regression was used to assess if there were significant differences in: (1) cognitive impairment (≤18 vs. >18), and (2) functional profile (≤2 vs. >2); by sex, type of institution (RH or DC), age, marital status (married not married) and attended school controlling for confounding variables. The response value for model 1 was dichotomized as 0 if an older person has no or mild cognitive impairment (total MMSE > 18) and as 1 if an older adult has moderate or severe cognitive impairment (total MMSE ≤ 18). The response value for model 2 was dichotomized as 0 if an older person has no or mild functional problem (total ENCS ≤ 2) and as 1 if an older adult has moderate, severe or complete functional problem (total ENCS > 2). The variable selection process proposed by Hosmer and Lemeshow was used [34]. Fractional polynomials and LOWESS smoother were used to evaluate linearity between age and the log odds [31]. Significant interactions among variables and type of institution was also tested. Diagnostics for the goodness-of-fit and residuals analysis were also employed. To account for possible lack of independence between subjects from the same type institution (RH or DC), it was also used the logistic mixed model were the type of institution was used as a random effect. Due to the low variance of the random intercept (σ^2^_model_ = 0.274; σ^2^_model 2_ = 0.788) the results are similar for type of institution, and we do not present these results. Statistical analyses were carried out using R (version 4.0.4).

## 3. Results

### 3.1. Sociodemographic Characteristics

The global sample consisted of older adults between 65 and 101 years old, with a mean age of 85.79 (SD = 6.95). In the Residential Home (RH) group the mean of age is 86.36 years (SD = 6.86) and in the Day Center (DC) group is 83.07 years (SD = 6.76). The majority of the participants are older than 85 years (60.6%). Most of the participants were female (69.6%), 66.4% were widowers and only 63.3% went to school (Table 1).

### 3.2. Functioning and Cognition

In the RH group, 56.6% of the older adults had cognitive impairment and in the DC group 39.8% had cognitive impairment. In DC group the majority has no cognitive deterioration, in RH the opposite is true. Figure 1 showed the MMSE total score for the two groups. The scores of older adults in the DC group are more similar to each other and higher than those recorded among older adults in RH. In addition, there is a significant difference between the scores of the DC group (Median = 23.0) and RH group (Median = 18.5), Mann–Whitney U test: W = 17389, *p* < 0.001).

Figure 2 presents the distribution of the results of the functional profile of the sample. The highest levels of dependence are found in “Self-Care” and “Learning and Mental Functions”. Dependency levels are significantly higher in RH group when compared to the DC group (Mann–Whitney U test: in all domains *p* < 0.001). In RH group the “Communication” dimension also reaches high dependency values.

There is a significant relation between group and the severity level (Chi-square test of independence: X2(4) = 56.763, *p* < 0.001). There are a higher percentage of adults with no problem or a mild problem and a lower percentage of moderate to complete problem in the DC group than in the RH group (Table 2). In this last group 58.9% of the older adults had moderate to complete problems and in the DC group this percentage decrease to 19.6%. These older adults need daily care in the activities of daily living.

Table 3 shows the percentage of RH group and DC group by level of severity of the general functioning profile, taking into account the variables sex, age group, marital status and educational level. In all variables evaluated, the most participants of both groups (RH and DC) had mild or moderate problem in functioning.

Univariate logistic regression analysis identified gender, school attendance, type of institution and age as relevant isolated factors for moderate or severe cognitive impairment and for moderate, severe or complete functional problem (Table 4 and Table 5).

In the multivariate logistic regression analysis, all these variables were identified to be correlated with the severity of cognitive impairment (Table 4). Women are about 1.6 more likely to have moderate or severe cognitive impairment than men. Older adults who attended school are 64% less likely to have a higher degree of cognitive impairment than those who did not attend school. The older adults in DC are 67% less likely to have a total MMSE ≤ 18 than those in RH. The chances of an older adults with moderate or severe cognitive impairment increase with age.

In the multivariate logistic regression analysis, age was not identified as being correlated with the severity of the functional problem (Table 5). Women are almost 1.6 more likely to have moderate, severe or complete functional problem than men. The older adults who attended school are 36% less likely to have a higher degree of functional problem than those who did not attended school. The older adults in DC are 84% less likely to have a total ENCS > 2 than those in RH.

## 4. Discussion

The functional profile of each person is interrelated with his or her sociodemographic context and with the biological, cultural and environmental characteristics of the individual, which was generally observed in the present study and is in line with the findings of other authors [31,35,36].

Older adults living at home had better scores than institutionalized individuals in all domains of ENCS and MMSE. The results remain stable when we insert other variables such as age, gender, marital status, and education, with the RH group with worse functioning and cognitive outcomes than the DC group. In a study conducted in Turkey comparing institutionalized and noninstitutionalized older adults, the authors concluded that institutionalized older adults had greater physical and social disability and a higher risk of psychiatric disorders, particularly depression, than those who lived in their homes [37]. Assuming that functioning is associated with quality of life, it should be noted that in a recent systematic literature review and meta-analysis, the authors concluded that older adults who had been institutionalized had a worse quality of life than those who lived at home [38]. In addition, according to other authors, functional and cognitive impairment, as well as lack of support and assistance with activities of daily living for the older adults, were considered predictive factors for institutionalization [39,40,41]. Additionally, a recent longitudinal study concluded that a lack of social support, living alone, nonparticipation in recreational and social activities, and not visiting family or friends were strong predictors of institutionalization among older adults [42]. In fact, in the present study we also concluded that the social relationships domain, where social network support is evaluated, has more deficits in RH group.

The sample included a greater number of women than men, a phenomenon that was called the “feminization of old age” in the scientific literature [43] and was reported by the World Health Organization [44]. Despite women live longer than men, these extra years are not usually lived in good health, as they have poorer health throughout their lives and higher poverty rates [45]. One of the factors contributing to this is that the role of caregiver is often attributed to women when a family member needs care. This role is unpaid and may limit their participation in the paid employment or education. However, this situation can have a significant cost on their own well-being in older age, increasing the risk of poverty, and in turn, decreasing well-being and quality of life [45]. The percentage of women in our study was higher than the percentage of women over 65 years in Portugal (58.2%) [46]. Therefore, this data indicated that women attend RH and DC more than men, perhaps due to the fact that they have a worse functional profile and more cognitive degradation than men, and consequently higher functional dependence. These data are in line with other studies that indicate that women have higher rates of functional dependence [47,48,49] and cognitive deficit [47]. This scenario can be interpreted by greater multimorbidity in older women than in men [13], along with the fact that women have a higher average life expectancy [50]. However, regarding functioning, when we analyze the variables sex, age, level of education and group (RH or DC), we found that functionality is worse in women, RH group and no attended school, regardless of age. This leads us to believe that the age factor is not a determining factor for worse functioning, so it is important to intervene on modifiable variables such as literacy and disease prevention and avoid institutionalization by keeping people at home.

In relation to school, a study conducted with the Brazilian population, illiterate individuals were more dependent for instrumental activities of daily living [51]). In addition, since functioning is associated with well-being and health and because higher levels of education are related to better physical or mental health, the results obtained in other studies [35,52,53] are also aligned with those of the present study. Additionally, multimorbidity is higher in people with lower educational levels [13]. In a contextual analysis of the current Portuguese reality, older adults who have not had the opportunity to attend school are those who had greater socioeconomic difficulties during childhood, and, for the most part, throughout life—the women often were removed from the family as children to work as maids in the homes of more affluent families until adulthood, and the men worked in low-paying jobs (when there were jobs available) and began to work at an early age. Low socioeconomic status combined with low literacy influences lifestyles throughout the life cycle. A recent study concluded that a healthy lifestyle benefits physical, psychological, cognitive and social functioning until old age [54], which in turn, is related to functioning and, consequently, to self-care. This, therefore, is a possible contextual explanation for the levels of dependence of illiterate people during the aging process. It is, therefore, necessary to promote opportunities for lifelong learning and the development of skills so that an active and healthy life is possible [55].

Regarding marital status, the analysis did not find that this sociodemographic variable had any association, either on the ENCS or on the MMSE. In fact, the literature points to the importance of the quality of social relationships, stating that the existence of bonds does not guarantee the availability of supportive social resources [3]. This means that being married or having children does not necessarily mean that social relationships are satisfactory and of quality or that this network offers support for activities of daily living or for instrumental activities of daily living or promotes communication or mental functions. However, society has undergone extensive changes in the family circle with the entry of women into the labor market, which limits their availability for the socially assigned role of family caregiver. Additionally, job opportunities are often located far from where parents live, so many older adults today live far from their children, which limits the availability of a close support network. Regarding spouses, as a rule, they are similar in age to one another, which may limit their ability to provide support in activities of daily living when necessary; these are factors that also contribute to institutionalization.

The results presented here add scientific knowledge to the limited existing literature and reinforce the importance of a person-centered care model. Patient-centered care is fundamental to self-care management and should guide care planning [56], i.e., we must take into account the person’s socio-demographic characteristics and the context in which he or she lives in order to plan the most appropriate care. Therefore, it is important that the institutions of older adults adopt care models based on the implementation of an individual care plan, which (i) is a person-centered tool; (ii) constitutes a space for dialogue among all caregivers; (iii) supports and facilitates route management; and (iv) integrates care services [57]. In addition, it is crucial that professionals pay attention to nonverbal communication in the care of older adults to improve the quality of life and cognitive function of them and their caregivers [58,59,60].

The limitations of the present study include its cross-sectional nature, which does not allow conclusions regarding causality.

## 5. Conclusions

We tested a new and important finding for aging policies, i.e., regardless of age, factors such as being a woman, not having gone to school, and being institutionalized are predictors of worse functioning. Therefore, it is important preserving functioning or slowing its loss and consequently promoting well-being and quality of life. For this purpose, a systematic evaluation of functioning is indispensable to ensure that care and contextual conditions are adapted to the health situation of the individual.

Although it is not possible to intervene directly in terms of age and sex, it is necessary to intervene by providing life-long training and implementing social and health responses that allow people to remain in their homes, because doing so affects the functioning of self-care and the quality of life of older adults. If it is not possible to proceed as described above, it is suggested to improve institutional responses so that they can promote the functioning of individuals of this age group. The model of providing care at home that is based on an individual care plan centered on the person and his/her close relationships and is associated with new artificial intelligence technologies for supporting older adults may be a relevant solution. Thus, it is suggested that experimental studies test this hypothesis.

We highlight and emphasize the importance of resilience and the ability to adapt to the new reality that is imposed on the current society. Thus, it is urgent to respond to this challenge with a model focused on self-care that prioritizes interpersonal relationships and the provision of care in the home.

## Figures and Tables

**Figure 1 ijerph-18-07030-f001:**
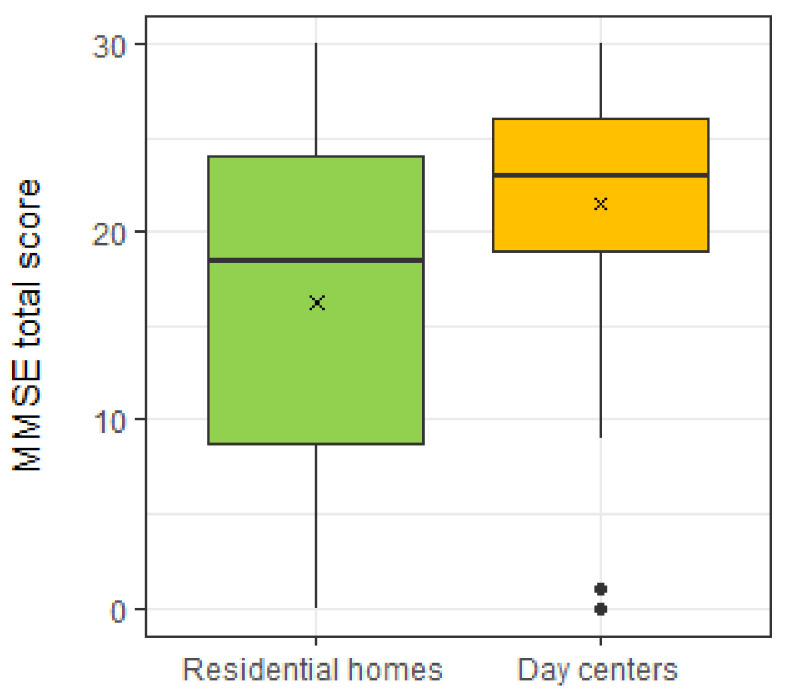
MMSE total score for two groups (RH e DC). The symbol x represents the mean and the black dots the outliers.

**Figure 2 ijerph-18-07030-f002:**
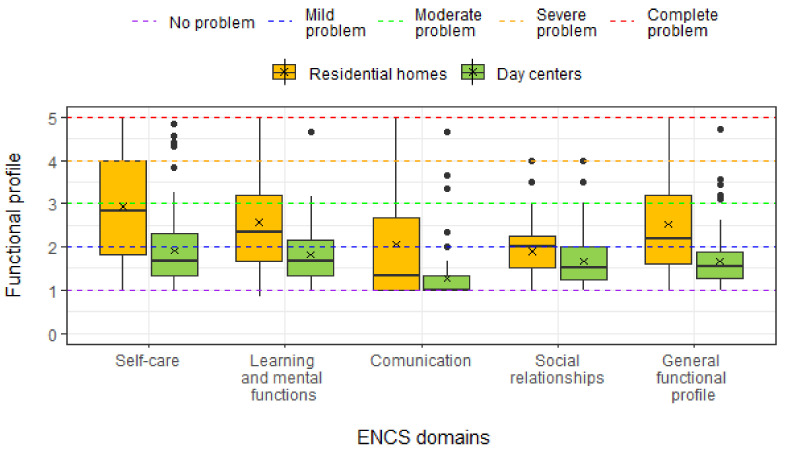
ENCS domain and general functional scores for two groups (RH e DC). The symbol x represents the mean and the black dots the outliers.

**Table 1 ijerph-18-07030-t001:** Characterization of the sample (N = 586).

	Total (N = 568)	RH (N = 484)	DC (N = 102)
Sex			
Female	408 (69.6%)	341 (70.5%)	67 (65.7%)
Male	178 (30.4%)	143 (29.5%)	35 (34.3%)
Age			
65–74	39 (6.7%)	28 (5.8%)	11 (10.9%)
75–84	192 (32.8%)	147 (30.4%)	45 (44.1%)
≥85	355 (60.6%)	309 (63.8%)	46 (45.1%)
Marital Status			
Single	66 (11.3%)	57 (11.8%)	9 (8.8%)
Married	106 (18.1%)	78 (16.1%)	28 (27.5%)
Widower	389 (66.4%)	331 (68.4%)	58 (56.9%)
Divorced	25 (4.3%)	18 (3.7%)	7 (6.9%)
Education			
Unlettered	188 (32.1%)	164 (33.9%)	24 (23.5%)
Did not go to school but knows how to read and write	27 (4.6%)	20 (4.1%)	7 (6.9%)
Attended school but not higher education	353 (60.2)	283 (58.5%)	70 (68.6%)
Higher education	18 (3.1%)	17 (3.5%)	1 (1.0%)

**Table 2 ijerph-18-07030-t002:** Functional profile assessed through the Elderly Nursing Core Set (N = 586).

	RH Group (N = 484)		DC Group (N = 102)	
General Functional Profile (Level of Severity)	N	%	N	%
No problem	35	7.2	15	14.7
Mild problem	164	33.9	68	66.7
Moderate problem	142	29.3	14	13.7
Severe problem	105	21.7	5	4.9
Complete problem	38	7.9	0	0

**Table 3 ijerph-18-07030-t003:** Participants’ sociodemographic characteristics and general functioning profiles (N = 586).

			MMSE		ENCS		
Variables		N	Without Cognitive Impairment (%)	Cognitive Impairment (%)	No Problem (%)	Mild or Moderate Problem (%)	Severe or Complete Problem (%)
Sex							
Female	RH	341	40.8	59.2	6.5	62.1	31.4
	DC	67	56.7	43.3	16.4	76.1	7.5
Male	RH	143	51.0	49.0	9.1	65.8	25.1
	DC	35	65.7	34.3	11.4	88.6	0
Age group, years							
65–74	RH	28	42.9	57.1	14.3	60.7	25.0
	DC	11	81.8	18.2	27.3	63.6	9.1
75–84	RH	147	44.9	55.1	6.1	61.2	32.7
	DC	45	66.7	33.3	17.8	80.0	2.2
≥85	RH	309	43.4	56.6	7.1	64.4	28.5
	DC	46	47.8	52.2	8.7	84.8	6.5
Marital Status							
Married	RH	78	50.0	50.0	10.3	60.2	29.5
	DC	28	60.7	39.3	7.1	92.9	0
Not married	RH	406	42.6	57.4	6.7	63.7	29.6
	DC	74	59.5	40.5	17.6	75.6	6.8
Attended school							
Yes	RH	283	45.6	54.4	7.8	67.9	24.3
	DC	70	60	40	15.7	82.9	1.4
No	RH	201	41.3	58.7	6.5	56.7	36.8
	DC	32	59.4	40.6	12.5	75.0	12.5

**Table 4 ijerph-18-07030-t004:** Univariate and multivariate logistic models for moderate or severe cognitive impairment (total MMSE ≤ 18).

	Univariate		Multivariate	
Variable	OR (95% CI)	*p* (Wald Test)	OR (95% CI)	*p* (Wald Test)
Sex (ref. male)	1.166 (1.129; 2.325)	0.009	1.648 (1.127; 2.423)	0.010
Attended school (ref. No)	0.343 (0.243; 0.482)	<0.001	0.360 (0.251; 0.512)	<0.001
Institution (ref. RH)	0.291 (0.174; 0.471)	<0.001	0.331 (0.194; 0.548)	<0.001
Age (in years)	1.045 (1.020; 1.071)	<0.001	1.027 (1.001; 1.054)	0.045
Marital Status (ref. Married)	1.529 (0.995; 2.374)	0.055		

Hosmer & Lemeshow Goodness-of-fit test: X2(7) = 9.150, *p* = 0.242. Nagelkerke R2 = 0.158. McFadden R2 = 0.091. Sensivity = 72.8%, specificity = 57.9, cut point = 0.423, AUC = 0.698.

**Table 5 ijerph-18-07030-t005:** Univariate and multivariate logistic models for moderate, severe or complete functional problem (total ENCS > 2).

	Univariate		Multivariate	
Variable	OR (95% CI)	*p* (Wald Test)	OR (95% CI)	*p* (Wald Test)
Gender (ref. male)	1.159 (1.118; 2.274)	0.012	1.590 (1.095; 2.313)	0.015
Attended school (ref. No)	0.611 (0.437; 0.853)	0.004	0.644 (0.452; 0.914)	0.014
Institution (ref. RH)	0.160 (0.092; 0.266)	<0.001	0.165 (0.094; 0.276)	<0.001
Age (in years)	1.033 (1.009; 1.058)	0.007		
Marital Status (ref. Married)	1.096 (0.719; 1.671)	0.669		

Hosmer & Lemeshow Goodness-of-fit test: X2(3) = 2.213, *p* = 0.530. Nagelkerke R2 = 0.149. McFadden R2 = 0.086. Sensivity = 70.1%, specificity = 54.6%, cut point = 0.574, AUC = 0.674.

## Data Availability

The data that support the findings of this study are available from http://35.181.116.247:4200/login (access date 4 January 2021), but restrictions apply to the availability of these data, which were used under license for the current study, and so are not publicly available. Data are however available from the authors upon reasonable request and with permission of University of Évora.

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
