# Peer review of "Functioning and Cognition of Portuguese Older Adults Attending in Residential Homes and Day Centers: A Comparative Study"

_ijerph, 2021, doi:10.3390/ijerph18137030_

Round 1
Reviewer 1 Report
This is a quite interesting study in older adults within the European Union, with clear implications for the quality of life in this increasing population, especially for the modifiable factors such as education and encouraging the improvement of social networks in the population.
Still I would suggest some minor adjustements:
- Introduction,
- Lines 91-99: I would also stress more the importance of international comorbidity education to aim for a better management of this increasing problem. An example could be:
- Lawson C, Pati S, Green J, Messina G, Strömberg A, Nante N, Golinelli D, Verzuri A, White S, Jaarsma T, Walsh P, Lonsdale P, Kadam UT. Development of an international comorbidity education framework. Nurse Educ Today. 2017 Aug;55:82-89. doi: 10.1016/j.nedt.2017.05.011. Epub 2017 May 17. PMID: 28535380.
- Materials and Methods,
- Lines 156-157, probably the symbol “£” is missing
- Results,
- Lines 172-173, probably DP stands for Standard Deviation? (maybe it was “Desvio padrão” in original?)
- Paragraph 3.1, Lines 171-177 I would suggest to add another table to synthesize these data to improve the ease in reading the numbers
- Line 184, I would suggest to write clearly if it is the median for the scores of the RH group and the DC group, instead of the acronyms “MERH” and “MeDC”
- Discussion
- Lines 260-261, I find the passage “So, this data indicate that women are more frequent of social responses” is a little unclear, maybe a rephrasing would be needed for a better understanding
- Lines 91-99: I would also stress more the importance of international comorbidity education to aim for a better management of this increasing problem. An example could be:
Author Response
Thank you very much for your comments. They were a great help in improving the manuscript. Below we respond to your comments.
Introduction: At the end of the paragraph we added a sentence to stress this importance.
Materials and Methods: It was added the correct symbol.
Results: Lines 172-173: It was corrected to SD; Paragraph 3.1, Lines 171-177: It was added a new table (Table 1); Line 184 - Instead of the acronyms we write clearly “median”.
Discussion: For a better understanding, we rephrase this part of the text (lines 264-270).
Reviewer 2 Report
Thank you for an opportunity to read and review this paper. The authors explore a very important, urgent and challenging phenomenon, which is the functioning and cognition of older people, and examine how these aspects are influenced by various care contexts. They point out that older women, who did not attend school, tend to have a higher degree of cognitive impairment than older men, who went to school. The authors also explore an important issue, which is ‘feminization of old age”.
They might also want to address the role of gender of care providers for older people. In Portugal (and, actually, on a global scale), the majority of carers are women, especially women of colour, immigrants or ethnic minorities with limited resources. In fact, it is socially assumed that women are the best caregivers since they are ‘naturally’ inclined to provide care, which reveals that care per se is feminized and generalized as female. Presenting carework as natural for women contributes to further marginalization and exploitation of often-unskilled women workers. Also, even it is not the focus of the paper, commenting briefly on the psychological well-being of caregivers themselves would also benefit the manuscript.
The authors conclude by suggesting interventions related to life-long training, social and health models for older people, and literacy and disease prevention in order to avoid institutionalization of older people and, instead, keep them at home. 
They also point out to the importance of a person-centered care model. The authors might find this recent article (Gerontologist. 2021 May 18: gnab066. doi: 10.1093/geront/gnab066) of their interest, which may help benefit the paper in terms the above mentioned aspects, and, especially, in regards to the importance of more person-centered care models in later life.
Also, please check these sentences:
Intro 25-6: ‘Women, older adults [older women] who did not attend school and those [who] live in Residential Homes…’
Lines 247-8: ‘the authors concluded that older 
adults [who had been] institutionalized had a worse quality of life than those who lived at home…’
Thank you and good luck with your further research.
Author Response
Thank you very much for your comments. They were a great help in improving the manuscript. Below we respond to your comments.
We have added your suggestion to the discussion: "Despite women live longer than men, these extra years are not usually lived in good health, as they have poorer health throughout their lives and higher poverty rates [45]. One of the factors contributing to this is that the role of caregiver is often attributed to women when a family member needs care. This role is unpaid and may limit their participation in the paid employment or education. However, this situation can have a significant cost on their own well-being in older age, increasing the risk of poverty, and in turn, decreasing well-being and quality of life [45]. "
It was added a sentence about work that you suggested (lines 324-326).
Line 25-26: The result is about all older adults and not just older women. In order to clarify, we rephrase the sentence to “Older women, older adults who did not attend school, and older adults who live in Residential Homes…”
Lines 247-8: Changed as suggested.
Thank you very much for your revision.